# Quantifying CanESM5 and EAMv1 sensitivities to Mt. Pinatubo volcanic forcing for the CMIP6 historical experiment

Landon A Rieger[1], Jason NS Cole[2], John C Fyfe[2], Stephen Po-Chedley[3], Philip J Cameron-Smith[3], Paul J Durack[3], Nathan P Gillett[2], and Qi Tang[3]

[1]Institute of Space and Atmospheric Studies, University of Saskatchewan, Saskatoon, SK
[2]Canadian Centre for Climate Modelling and Analysis, Environment Canada, Victoria, BC
[3]Lawrence Livermore National Laboratory, Livermore, California

**Correspondence:** Landon Rieger (landon.rieger@usask.ca)

**Abstract.** Large volcanic eruptions reaching the stratosphere have caused marked perturbations to the global climate including cooling at the Earth's surface, changes in large-scale circulation and precipitation patterns and marked temporary reductions in global ocean heat content. Many studies have investigated these effects using climate models, however uncertainties remain in the modelled response to these eruptions. This is due in part to the diversity of forcing datasets that are used to prescribe the distribution of stratospheric aerosols resulting from these volcanic eruptions, as well as uncertainties in optical property derivations from these datasets. To improve this situation for CMIP6 a two step process was undertaken. First, a combined stratospheric aerosol dataset, the Global Space-based Stratospheric Aerosol Climatology, GloSSAC (1979-2016), was constructed. Next, GloSSAC, along with information from ice-cores and sun photometers, was used to generate aerosol distributions, characteristics and optical properties to construct a more consistent stratospheric aerosol forcing dataset for models participating in CMIP6. This "version 3" of the stratospheric aerosol forcing has been endorsed for use in all contributing CMIP6 simulations. Recent updates to the underlying GloSSAC from version 1 to version 1.1 affected the 1991 to 1994 period and necessitated an update to the stratospheric aerosol forcing from version 3 to version 4. As version 3 remains the official CMIP6 input, quantification of the impact on radiative forcing and climate is both relevant and timely for interpreting results from experiments such as the CMIP6 historical simulations. This study uses two models, the Canadian Earth System Model version 5 (CanESM5), and Energy Exascale Earth System Model (E3SM) Atmosphere Model version 1 (EAMv1) to estimate the difference in instantaneous radiative forcing in simulated post-Pinatubo climate response when using version 4 instead of version 3. Differences in temperature, precipitation, and radiative forcings are generally found to be small compared to internal variability. An exception to this is differences in monthly temperature anomalies near 24 km altitude in the tropics, which can be as large as 3°C following the eruption of Mt. Pinatubo.

## 1 Introduction

The stratosphere holds a layer of aerosols consisting primarily of sulfuric acid and water that impact climate in a variety of ways (Kremser et al., 2016). Most importantly, this stratospheric aerosol layer scatters incoming light leading to a surface cooling effect. Scattering in the atmosphere can be greatly enhanced by volcanic eruptions, which in turn strengthens this

surface cooling. For example, the 1991 eruption of Mt. Pinatubo injected an estimated 5-10 Tg of sulfur into the stratosphere (Guo et al., 2004; English et al., 2013; Dhomse et al., 2014; Timmreck et al., 2018), resulting in a peak top-of-atmosphere radiative forcing of roughly 3-4 Wm$^{-2}$ (Ramachandran et al., 2000; Hansen et al., 1992), and cooled global temperatures by a few tenths of a degree Celsius (Robock and Mao, 1995; Thompson and Solomon, 2009). There was also a significant impact on

oceans, with global ocean heat content decreasing by $3 \times 10^{22}$ J, and sea level decreasing by 5 mm (Church et al., 2005). Over the last two decades a number of smaller volcanic eruptions have also injected sulfur dioxide into the stratosphere (Vernier et al., 2011). These eruptions have had a small but discernible effect on global temperature (Solomon et al., 2011; Fyfe et al., 2013; Santer et al., 2014; Schmidt et al., 2018; Stocker et al., 2019).

    Issues remain in characterizing the radiative forcing caused by changes in stratospheric aerosols and the climate responses

that result. General circulation models, for example, often overestimate the stratospheric warming response following the eruption of Mount Pinatubo (Lanzante and Free, 2008; Gettelman et al., 2010). The impact of smaller, post-Pinatubo volcanic eruptions on surface cooling needs better quantification (Santer et al., 2014), and large uncertainties remain in characterizing the response of the upper troposphere and lower stratosphere to volcanic eruptions (Ridley et al., 2014; Andersson et al., 2015). Such uncertainties in the climate response to volcanic eruptions result in part from differences in the stratospheric

aerosol datasets that are prescribed in general circulation models (GCMs). GCMs participating in the Fifth Coupled Model Intercomparison Project (CMIP5) (Taylor et al., 2012; Driscoll et al., 2012) used stratospheric aerosol datasets from Ammann et al. (2003) and Sato et al. (1993), while post-CMIP5 simulations (Solomon et al., 2011; Fyfe et al., 2013) have used datasets that include recent eruptions (Vernier et al., 2011). To avoid a diversity of stratospheric aerosol datasets, and their associated forcings, a homogenized stratospheric aerosol time series was developed for use with CMIP6 (Durack et al., 2018).

This stratospheric aerosol dataset will help reduce uncertainties that have resulted from differing stratospheric aerosol assumptions (e.g., in CMIP5). While modelling centres were performing simulations for CMIP6 an update was made to this dataset that affected the stratospheric aerosol loading of the Pinatubo eruption. Although the updates are not CMIP6 endorsed, and forcing should remain consistent across models, the changes to aerosol loading can be substantial. To estimate the potential effect of these changes on CMIP6 results we characterize the impact of this dataset update on the global climate using two

general circulation models.

## 2   The CMIP6 Stratospheric Aerosol Dataset

For CMIP6 experiments the stratospheric aerosol forcing dataset post-1980 was constructed in a two step process. First, data from multiple satellite instruments was compiled into a continuous extinction record at 525 nm that spans the entire period. Extinction measurements at other visible and near-infrared wavelengths are available for portions of the record, but do not

span its entirety. This composes the Global Satellite-based Stratospheric Aerosol Climatology, or GloSSAC (Thomason et al., 2018). The second step is deriving the asymmetry factor, single scattering albedo and extinction at the wavelengths required for radiative transfer calculations in climate models (Luo, 2018a). This was done by deriving a particle size distribution from the measurement periods where multiple wavelengths are available (1985-2005) and extrapolating to periods where they are not

(pre-1985 and post-2005). This composes the IACETH-SAGE3lambda-3-0-0 dataset available from input4MIPs. Luo (2018a) then used the particle size distributions to compute the optical parameters at the wavelength bands of participating models. In this way, the optical properties required for each participating model's radiative transfer scheme are consistent with the underlying extinction and particle size climatology. As of May 31, 2016 these were available from `ftp://iacftp.ethz.ch/pub_read/luo/CMIP6` and represented version 3 stratospheric aerosol datasets used for CMIP6, herein referred to as version 3, or 'v3' for brevity.

After publication of these datasets an error was found in the GloSSAC processing involving the cloud clearing in the CLAES data, necessitating an update in GloSSAC from version 1 to version 1.1, and subsequent update in the CMIP6 forcing dataset from version 3 to version 4 (Luo, 2018b). The update was published August 27, 2018 and is available from `ftp://iacftp.ethz.ch/pub_read/luo/CMIP6_SAD_radForcing_v4.0.0`. As it is not yet officially CMIP endorsed it is not currently available from input4MIPs. Changes to the data processing primarily affected only a subset of the satellite measurements which contributed to the January 1991 to December 1994 period, so data outside of this range remains the same between version 3 and version 4.

Of most direct consequence to GCM experiments is the stratospheric aerosol optical depth (SAOD), calculated as the vertical integral of aerosol extinction from the tropopause to the top of the atmosphere, or in this case to the top of the dataset at 40 km. Luo (2018b) show the magnitude of these changes at several latitude bins and times for 1020 nm, and the following analysis expands on this at 550 nm in the context of this paper. The top panel in Figure 1 shows the monthly SAOD at 550 nm from the version 3 dataset. The impact of the Mount Pinatubo eruption on SAOD is largest in the tropics, but a substantial amount of aerosol is transported to the extra-tropics in both hemispheres in ensuing months, and global SAOD values remain elevated over background for several years. The bottom panel shows the SAOD differences between version 3 and version 4. The peak Pinatubo-induced SAOD values are smaller in version 4, but v4 exhibits modest increases at other locations, reducing globally averaged differences and potentially offsetting climate differences between versions. Fractional differences show a very similar pattern with differences in the thickest part of the plume reaching 20%, and differences nearer the poles reaching 50% for short periods of time.

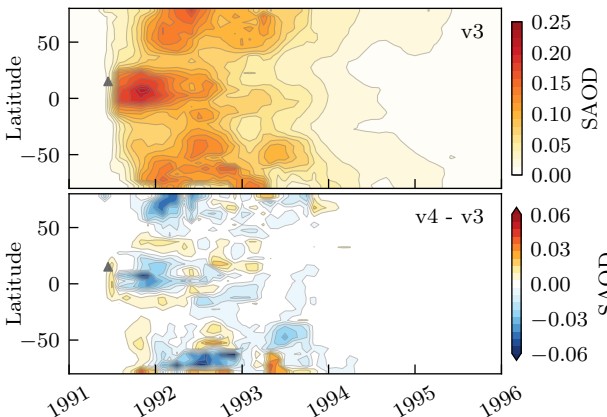

**Figure 1.** The top panel shows aerosol optical depth in the stratosphere (SAOD) at 550 nm from the v3 dataset. The bottom panel shows the absolute difference between the versions, computed as v4 − v3, during this same time period. The triangle marks the Pinatubo eruption at 14°N on June 15, 1991.

However, when looking at SAOD alone, decreases in extinction at higher altitudes can be offset by increases nearer the tropopause, reducing the apparent differences. This can be seen in Figure 2, which shows the difference in extinction at 550 nm as a function of altitude and time for the global average and three latitude bands that showed large changes in SAOD. The most prominent differences between v3 and v4 are evident in the tropics where the main aerosol plume has been reduced in optical thickness by up to 50%. Conversely, altitudes below the main plume have increased extinction. While the increase extends to the ground, most of these altitudes are below the tropopause, and so are not considered in climate simulations (CMIP6 simulations are recommended to use stratospheric aerosols only above the tropopause (Thomason et al., 2018)). The solid gray lines in Figure 2 indicate the monthly averaged thermal tropopause from NCEP1 reanalysis data (Kalnay et al., 1996).

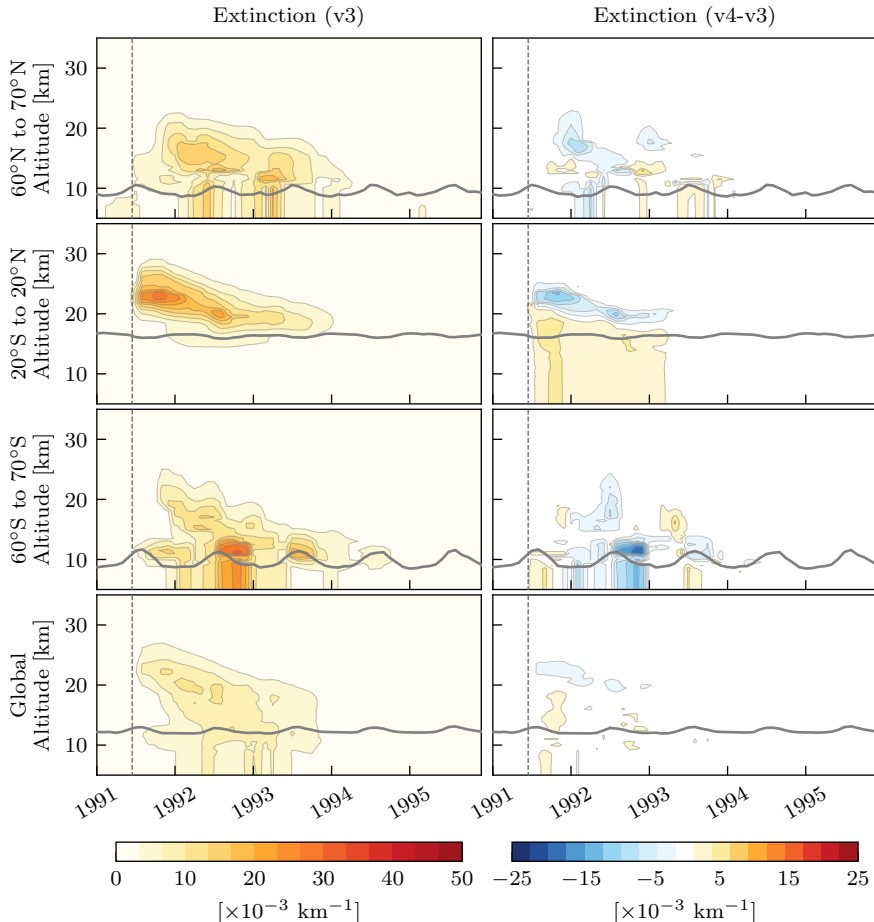

**Figure 2.** The left column shows the version 3 extinction at 550 nm in four latitude bands. The right column shows the change in extinction when the newer version 4 data product is used (v4-v3). The solid gray lines show the monthly NCEP1 thermal tropopause and the dashed line indicates the date of the Pinatubo eruption.

It is worth highlighting here that since the creation of the version 4 stratospheric aerosol dataset updates to the underlying GloSSAC dataset have continued, with version 2 now available (Kovilakam et al., 2020). Figure S1 shows the changes in stratospheric aerosol optical depth between GloSSAC versions 1, 1.1 and 2. While these recent updates to GloSSAC have not been incorporated into the CMIP6 stratospheric aerosol forcing dataset, the magnitude of the differences between GloSSAC

5    version 1 and version 2 are at least as large in magnitude (although very different in distribution) to the differences between version 1 and version 1.1 of GloSSAC (Aubry et al., 2020). This makes it likely that future updates to stratospheric aerosol forcing will result in climate effects of comparable magnitude to those seen in this study.

**Table 1.** Models and model configurations used in this analysis.

| Experiment | Model | Mode | Ensemble size (v3/v4) |
|---|---|---|---|
| Radiative Forcing | CanESM5 | AMIP | 2 (1/1) |
| Temperature | CanESM5 | ESM | 30 (15/15) |
|  | EAMv1 | AMIP | 6 (3/3) |
| Ocean Heat Content | CanESM5 | ESM | (15/15) |

## 3   Experimental Setup

This paper looks at three types of impacts caused by changes to the stratospheric aerosol forcing; from the immediate radiative and heating differences, to short term temperature effects, to longer-term changes represented by ocean heat content. Table 1 shows the three experiments, the models used for each, and whether they were ran in atmosphere-ocean coupled mode (ESM) or uncoupled with prescribed ocean temperatures (AMIP). The analysis uses two models, the Canadian Earth System Model version 5 (CanESM5) which has a relatively coarse horizontal resolution that enables simulation of large ensembles in a fully coupled ocean-atmosphere mode, and the E3SM Atmosphere Model version 1 (EAMv1) which is used to verify effects in higher resolution models, albeit with a smaller ensemble.

Simulations for examining impacts on instantaneous radiative forcing and heating rates were performed with CanESM5 in atmosphere-only mode using sea-surface temperature and sea-ice prescribed by observations for the period 1990-1999 following the AMIP protocol (Gates et al., 1999). For this analysis one simulation spanning from 1989 to 2014 was performed with the version 3 forcing data and a second with the version 4. Since only differences in instantaneous quantities are explored in this analysis a larger ensemble was not performed.

To explore the climate response to volcanic forcing, transient historical experiments were performed with CanESM5 following the methodologies set forth for CMIP6 (Eyring et al., 2016) using the version 3 aerosol climatology and all standard forcings. Simulations using version 4 of the stratospheric aerosol data were performed with the same protocol by branching new simulations off of CanESM5 historical simulations at the end of 1989. This was done for 15 realizations, with simulations using version 4 data run until 2014. In addition, three realizations were performed using the EAMv1 model. As with the CanESM5 simulations, we consider both version 3 and 4 of the volcanic aerosol dataset, performing three simulations with each version, using the CMIP6 protocols for the period 1990-1999. The reason for not using the coupled E3SM, and the reduction in ensemble size is the considerably higher computational cost of E3SM and EAMv1 compared to CanESM5. Additionally, as explained later, using coupled and uncoupled CanESM5 simulations yield very similar results for our analysis, indicating the use of a fully coupled model is not necessary.

## 3.1 CanESM5

We provide here a brief description of CanESM5 but a more thorough overview of the components and properties of CanESM5 is given in Swart et al. (2019). The atmospheric component of CanESM5, the Canadian Atmospheric Model version 5 (CanAM5) is a spectral model employing T63 triangular truncation with physical tendencies calculated on a 128×64 (~2.81°
or approximately 300 km at the equator) horizontal linear grid. CanAM5 has 49 unevenly spaced vertical levels up to ~0.1 hPa, with a vertical resolution of approximately 1.5 km near 25 km altitude. The physical ocean component of CanESM5 is based on NEMO version 3.4.1 (Madec and Imbard, 2012) and has 45 levels with approximately 6 m resolution in the upper ocean increasing to ~250 m in the lower ocean, and a horizontal resolution of approximately 1°. CanAM5 does not simulate a QBO.

## 3.2 EAMv1

In addition to CanESM5, we also consider simulations from version 1.0 of the Energy Exascale Earth System Model (E3SM) (Golaz et al., 2019). In particular, we employ the E3SM Atmosphere Model version 1 (EAMv1) (Rasch et al., 2019) using prescribed sea surface temperature (SST) and sea ice concentrations as boundary conditions. SST and sea ice fields are from HadISST version 1.1.3 (Durack and Taylor, 2017) as described in Hurrell et al. (2008). The model solves the atmospheric primitive equations using a continuous Galerkin spectral finite element method and has a horizontal resolution of approximately
100 km (or 1°) at the equator. EAMv1 has 72 unevenly spaced vertical levels with a model top at ~0.1 hPa, and vertical resolution of approximately 1-2 km in the lower stratosphere. EAMv1 employs a linearized ozone chemistry scheme (Hsu and Prather, 2009). In the realizations used here, EAMv1 produces a quasi-biennial oscillation (QBO) that tends to be too frequent compared to observation, but does sample a variety of states during the eruption period. While the frequency and strength of the QBO can be greatly improved by modifying parameterized convectively generated gravity waves (Richter et al., 2019), the
improvement is not included in the current simulations.

## 4 Results

### 4.1 Radiative Forcing

Volcanic aerosols absorb near-infrared and thermal radiation, heating the stratosphere, while simultaneously cooling the troposphere due to scattering of visible and near-infrared radiation. The magnitude of this effect has been investigated in numerous
studies, with marked decrease in radiation at the surface (Dutton and Christy, 1992) and tropopause (Hansen et al., 1992), and increases in reflected radiation at the top of the atmosphere (Minnis et al., 1993; Stenchikov et al., 1998; Ramachandran et al., 2000). For this work, radiative forcing is computed as the instantaneous net incoming flux at the top of the atmosphere for solar wavelengths (less than 4 microns) and thermal wavelengths (greater than 4 microns). Net total radiative forcing is calculated as the sum of the solar and thermal components.
For the CanESM5 AMIP used here an additional, diagnostic, atmospheric radiative transfer calculation is performed in which the stratospheric aerosol is zeroed out. The difference between the two computations gives the instantaneous radiative forcing

due to the presence of the stratospheric aerosols. Figure 3 shows the solar, thermal and net stratospheric aerosol radiative forcings from the version 3 and version 4 aerosol datasets for the global average and three latitude bands. The thermal forcing from the Pinatubo eruption peaks at approximately $2.5\,\mathrm{Wm^{-2}}$ in the tropics and $1.5\,\mathrm{Wm^{-2}}$ globally. Solar forcing is larger, peaking at nearly -6 and $-3.7\,\mathrm{Wm^{-2}}$ for the tropics and global averages, respectively. Polar regions can also have large solar forcing during the summer months, but the smaller geographic area and seasonal cycle due to reduced solar insolation in the winter lead to only a small impact globally. Differences in radiative forcing between version 3 and 4, shown as the shaded gray region, are much smaller, with net forcing differences peaking at $0.44\,\mathrm{Wm^{-2}}$ in the tropics and $0.11\,\mathrm{Wm^{-2}}$ globally. While maximum forcing differences in the SH polar regions are larger than the tropics (up to $1\,\mathrm{Wm^{-2}}$), the relatively small area contributes little to the global averages.

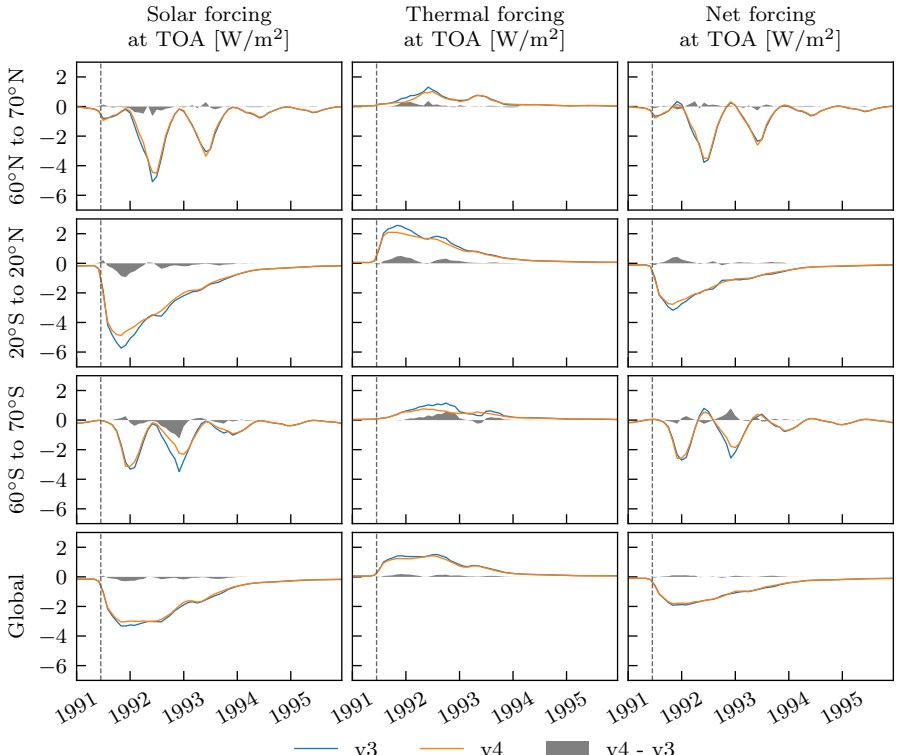

**Figure 3.** The left column shows the instantaneous solar radiative forcing at the top of atmosphere due to stratospheric aerosols in CanESM5. The blue line shows the forcing from version 3, the orange from version 4, and the gray shaded region indicates the difference in forcing between the two datasets (v4 -v3). The center column shows the instantaneous thermal radiative forcing at top of atmosphere, and the right column shows the instantaneous net radiative forcing at top of atmosphere. Each row shows the results for a different latitude band.

The absorption of both solar and thermal radiation heats the stratosphere in regions of enhanced aerosol loading (Kinne et al., 1992; Kinnison et al., 1994; Stenchikov et al., 1998; Andersen et al., 2001). Figure 4 shows the vertical distribution of instantaneous heating rates due to stratospheric aerosols computed using version 3 of the aerosol dataset. The largest heating

rate of 0.5°C/day in the tropics near 24 km is seen approximately 6 months after the eruption, and contributes to a globally averaged instantaneous heating rate of 0.2°C/day, with increases over background that last until early 1994. The differences in instantaneous heating rates from version 3 to version 4 are also shown in Figure 4. At times, heating rates have been reduced in version 4 by almost half where extinction is largest, with slight increases in heating rates closer to the tropopause. This result

5   is consistent with the differences between the aerosol datasets (Figure 2).

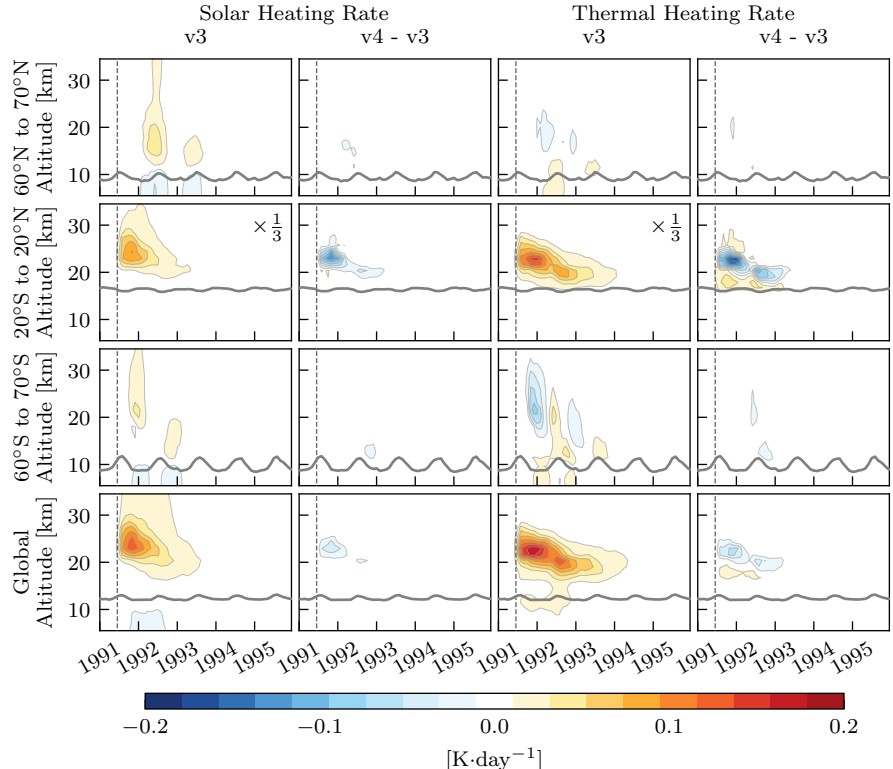

**Figure 4.** Instantaneous heating rates from CanESM5 model runs. The left column shows the instantaneous solar heating rate due to stratospheric aerosols using the version 3 forcing dataset. The second column shows the change in the instantaneous solar heating rate when version 4 is used instead. The third and fourth columns show the same, expect for the instantaneous thermal heating rates. The gray lines denote the tropopause, and dashed lines indicate the date of the Pinatubo eruption. Note that the v3 instantaneous heating rates in the tropics have been multiplied by 1/3 for visual representation.

## 4.2   Climate Response

The radiative heating rates induced by volcanic eruptions translate to substantial warming in tropical stratospheric temperature anomalies in CMIP5 models, ranging from 2°C to nearly 10°C in the tropical stratosphere (Douglass and Knox, 2005; Driscoll et al., 2012; Arfeuille et al., 2013).

The ensemble mean CanESM5 stratospheric temperature anomalies following the Pinatubo eruption are shown in the left column of Figure 5. Temperature anomalies up to 7°C are seen in the tropics near 24 km, where the extinction, and solar heating rates are largest. The stratosphere also exhibits a long-term stratospheric cooling, but linearly detrending the record still results in a maximum temperature anomaly of 6°C, and does not change differences seen between version 3 and version 4. The

structure of the global averaged temperature anomaly largely follows the evolution of the tropical temperatures, but peaks at approximately 3°C, which results because temperature anomalies outside of the tropics are considerably smaller. When using version 4 of the stratospheric aerosol data the peak temperature anomalies in the tropics are reduced by just over 2°C, and peak global anomalies are reduced by 0.8°C. The stratospheric temperature response in the polar regions can differ by 2-4°C, depending on the version of the SAOD used, but these differences are considerably smaller than the between-realization vari-

ability, which has a standard deviation of 6°C in these regions. Increases in stratospheric temperature as modelled by EAMv1 are similar in magnitude (see Figure S2). However, with only three ensemble members, temperature differences compared to version 4 are not statistically significant at the 95% level.

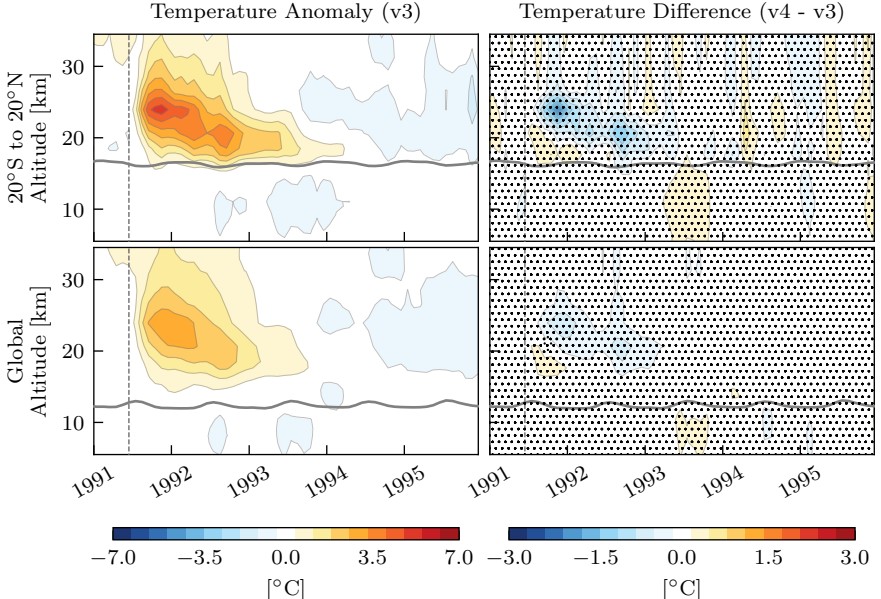

**Figure 5.** The left column shows the monthly temperature anomalies averaged over all CanESM5 ensemble members with version 3 forcing for 20°S to 20°N (top) and the global average (bottom). The right column shows the difference in temperature when using the version 3 and version 4 datasets. Stippling marks the regions where differences are not significantly different from zero at the 95% confidence level. Anomalies are computed using the simulation period of 1990 through 1999.

Figure 6 shows the same data as a function of latitude and time. The top panel indicates the maximum difference in temperature anomalies at any altitude, with the bottom panel indicating the altitude at which the difference occurs. No temperature

differences outside of approximately 30°S to 30°N are evident, nor are changes in the temperature gradient (not shown).

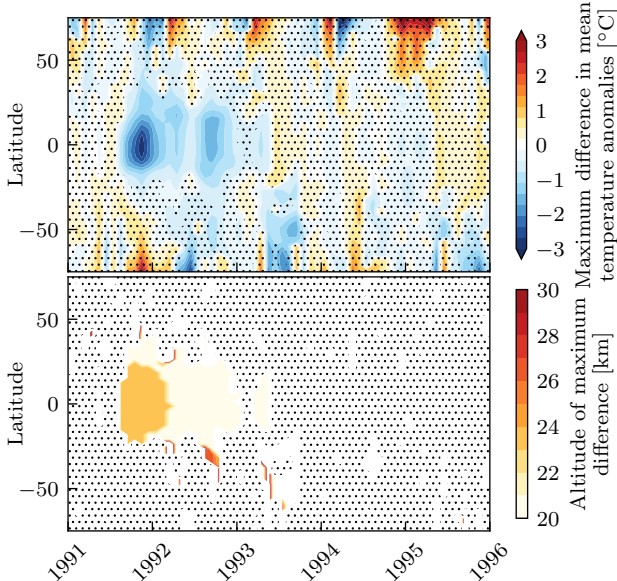

**Figure 6.** The top panel shows the maximum difference in monthly temperature anomalies at any altitude as a function of latitude and time. Stippling marks the regions where differences are not significantly different from zero at the 95% confidence level. The bottom panel shows the altitude at which the maximum occurs.

For observational comparisons data from the Remote Sensing Systems (RSS) microwave temperature record (Mears and Wentz, 2009; Mears and Wentz, 2009) is used. The RSS datasets are composed of measurements from the Microwave Sounding Units and Advanced Microwave Sounding Units which provide temperature information for several deep atmospheric layers. For this study, the temperature of the lower stratosphere (TLS) and temperature of the lower Troposphere (TLT) products are
5 used. TLS is a weighted average from approximately 10 to 30 km, with a peak sensitivity at 17.4 km and 75% of the contribution coming from between 14 to 22 km. TLT data is from below 10 km with over 75% of the signal from below 5.5 km including a 10-15% contribution from the surface (Mears and Wentz, 2017). Comparable temperature records from CanESM5 and EAMv1 are computed by applying RSS weighting functions to model temperatures. Since TLT includes a significant contribution from the surface, the weighting function depends on the surface type (land or ocean) to account for surface emissivity differences.
Figure 7 shows TLS and TLT anomalies simulated by CanESM5 and EAMv1 using version 3 and version 4 of the dataset in blue and orange respectively, as well as the RSS measurements in black. TLS anomalies are smaller in version 4 by up to 0.5°C in the tropics and 0.25°C globally, but modelled TLS values still remain approximately 1°C above measured values when globally averaged. This model-observational difference in stratospheric temperature has also been noted in other GCMs (Lanzante and Free, 2008; Gettelman et al., 2010). EAMv1 stratospheric temperatures in the tropics show increased variability
when compared to CanESM5 due to the quasi-biennial oscillation, which is not present in CanESM5. Tropospheric temperature differences arising from changes to the aerosol dataset are smaller than in the stratosphere for both models. EAMv1 results exhibit much smaller variability between ensemble members in the lower troposphere due to specified sea surface temperatures,

and do not show a significant difference between versions 3 and 4. CanESM5 differences in TLT of approximately 0.2°C in the tropics and 0.1°C globally are present in 1993, but result primarily from a change in the phase of the El Niño Southern Oscillation (ENSO) between version 3 and version 4 runs.

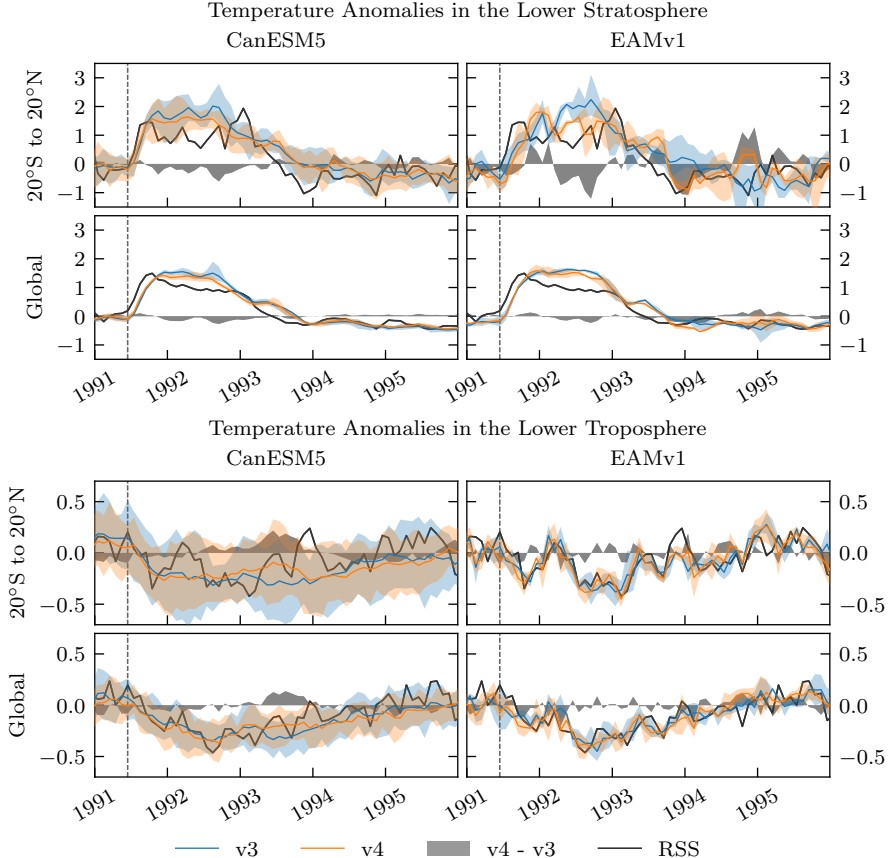

**Figure 7.** Temperature anomalies in the lower stratosphere (top two rows) and lower troposphere (bottom two rows) in the tropics and globally averaged. The left column shows results from CanESM5 and the right column EAMv1. Solid blue lines show the mean temperature anomaly using the version 3 dataset and orange show the same using version 4. The blue and orange shaded regions show the 10th and 90th percentiles for version 3 and 4 respectively for the CanESM5 ensemble and the max/min range for the EAMv1 ensemble. The gray region shows the mean difference between simulations using version 3 and version 4. The black line shows the RSS observations and the dashed line marks the eruption of Pinatubo on June 15th, 1991. Anomalies are computed using the simulation period of 1990 through 1999.

While large eruptions may increase the likelihood and magnitude of El Niño events (Adams et al., 2003; Mann et al., 2005; Emile-Geay et al., 2008; Khodri et al., 2017), the response is heavily model dependent (Predybaylo et al., 2017). Due to non-zero changes in the volcanic forcing before the eruption, the version 3 and version 4 ensembles are not in identical internal states on June 16th when Pinatubo erupts, with the version 4 ensemble tending slightly to more La Niña-like states. This can have a marked effect on the climate response (Lehner et al., 2016; Pausata et al., 2016; Zanchettin et al., 2019). However,

there is no apparent difference in ENSO states in the 2 years following the eruption when initial conditions are taken into account. Similar investigation of the North Atlantic Oscillation (NAO) shows no significant impact on the phase or magnitude. Additionally, changes to the aerosol forcing do not have a significant impact on ocean heat content, as shown in Figure 8. As such, changes to atmospheric temperatures in CanESM5 AMIP runs (not shown) show comparable changes to coupled runs, and therefore the AMIP runs from the EAMv1 model are expected to be a good representation of results from the coupled model.

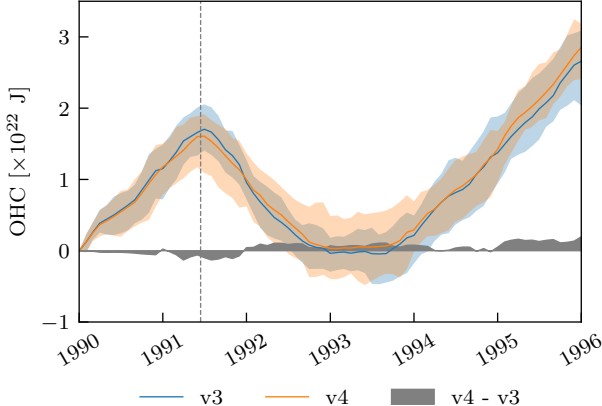

**Figure 8.** Monthly deseasonalized ocean heat content relative to January 1990. The blue line shows OHC using the version 3 aerosol forcing while the orange line indicates results when using version 4. The shading indicates the 10$^{\text{th}}$ and 90$^{\text{th}}$ percentiles of the data.

In addition to temperature responses, several studies have noted a decrease in global precipitation following the Pinatubo eruption (Robock and Liu, 1994; Broccoli et al., 2003; Gillett et al., 2004; Barnes et al., 2016). The CanESM5 ensemble mean shows a global precipitation decrease of 0.05 mm/day one year after the eruption, consistent with previous studies. However, this is similar for both the version 3 and version 4 datasets, and differences in precipitation due to the change in stratospheric aerosols are not statistically significant. These results are summarized in Figure 9 that shows the changes in climate response when using version 3 and version 4 aerosols averaged for two years following the Pinatubo eruption. For CanESM5, with results shown as the box and whisker plots, the largest differences in temperatures between aerosol datasets occur near 24 km and are statistically significant from zero at the 95% confidence level, as are the changes in the TLS. Changes in lower tropospheric temperatures are not statistically significant between versions, nor are changes in precipitation. EAMv1 results are shown as individual points, and indicate similar responses to those seen in CanESM5.

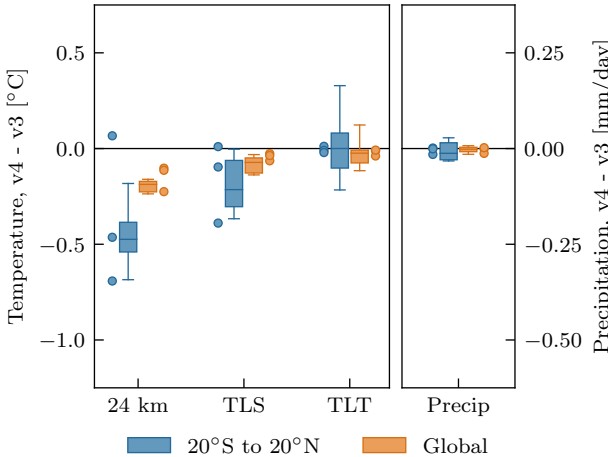

**Figure 9.** Difference in temperature and precipitation levels between version 3 and version 4 for the 2 years following the Pinatubo eruption. Boxes show the interquartile range of the CanESM5 data from the 15 member ensemble, and whiskers mark the 10[th] and 90[th] percentiles. Individual markers show the values from each of the three EAMv1 ensemble realizations.

## 5 Conclusions

Since publication of the version 3 stratospheric aerosol dataset recommended for CMIP6, updates included in version 4 have resulted in extinction differences as large 50% in the aerosol plume of the Pinatubo eruption. When these datasets are used in CanESM5 it is found that using version 4 instead of version 3 caused reductions in the instantaneous top-of-atmosphere radiative fluxes up to $0.44 \, \text{W/m}^2$ in the tropics approximately 6 months following the eruption and maximum differences in instantaneous radiative heating rates of 0.2 °C/day in the tropics. The substantial change in stratospheric heating rates at specific altitudes following the eruption results in significant temperature response differences of up to 3°C. Over deeper layers and larger spatial scales the impact is less pronounced with only the TLS showing statistical significance. As a result, the impact on global precipitation rates is also small. Temperatures in the lower stratosphere, between approximately 14 and 22 km, are decreased by 0.2°C with no statistically significant change in the lower troposphere. Similarly, precipitation rates and changes to the ENSO index are not substantial enough to be distinguished from unforced internal model variability. Based on results from two models participating in CMIP6, we find that the impact of the update from version 3 to version 4 of the stratospheric aerosol dataset is relatively small for the fields considered of radiative forcing, temperature, and precipitation. This indicates that while there is a known forcing issue in the v3 stratospheric aerosol dataset, this does not undermine the utility of the CMIP6 historical ensemble to quantify the anthropogenic-forced impact on the climate. Use of the new SAOD dataset may, however, affect quantities not considered in this study, and its impact may be model dependent, so modelling groups interested in the post-Pinatubo response may want to assess the impact of the new SAOD dataset in their models.

*Code and data availability.*   The CanESM5 model code is available at https://gitlab.com/cccma/canesm. CanESM5 model data and analysis code is available at 10.5281/zenodo.3524445. The E3SM project, code, simulation configurations, model output, and tools to work with the output are described at the E3SM website (https://e3sm.org). Instructions on how to get started running E3SM and its components are available at the E3SM website (https://e3sm.org/model/running-e3sm/e3sm-quick-start). All model codes may be accessed on the GitHub
repository (at https://github.com/E3SM-Project/E3SM)

*Author contributions.*   LR, JCN, JCF and NPG contributed to the manuscript writing, preparation and design of the CanESM5 experiments. SP contributed to the manuscript writing, preparation and design of the E3SM experiments. PJD, PJC-S and QT contributed to the design of E3SM experiments and reviewed the manuscript.

*Competing interests.*   The authors declare that they have no conflict of interest.

*Acknowledgements.*   This work was funded by the Canadian Space Agency through the Earth System Science Data Analyses project. Work performed by S. Po-Chedley, P. J. Cameron-Smith, P. J. Durack, and Q. Tang at Lawrence Livermore National Laboratory (LLNL) was under the auspices of the U.S. Department of Energy under Contract DE-AC52-07NA27344. S. Po-Chedley was funded by LLNL LDRD 18-ERD-054. The work of P. J. Cameron-Smith and Q. Tang was under the Energy Exascale Earth System Model (E3SM) project, which is funded by the U.S. Department of Energy, Office of Science, Office of Biological and Environmental Research. NCEP Reanalysis data provided by
the NOAA/OAR/ESRL PSD, Boulder, Colorado, USA, from their web site at https://www.esrl.noaa.gov/psd/. The authors would also like to thank Benjamin Santer, James Anstey and Adam Bourassa for comments on the manuscript and the development teams of E3SM and CanESM5.

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
