# Peer review of "Quantifying CanESM5 and EAMv1 sensitivities to Mt. Pinatubo volcanic forcing for the CMIP6 historical experiment"

_Geoscientific Model Development, 2019_

## Referee Comment (RC1) · Claudia Timmreck (Referee) · 5 Mar 2020

The paper describes the differences in instantaneous radiative forcing, temperature, and precipitation in simulated post-Pinatubo climate changes when using version 4 of the CMIP6 Stratospheric Aerosol data set instead of version 3 in the Canadian Earth System Model version 5 (CanESM5) and in the Energy Exascale Earth System Model (E3SM) Atmosphere Model version 1 (EAMv1). In general, the differences between both versions of the volcanic forcing data set are small compared to internal variability except for temperature anomalies in the tropical stratosphere. Overall, the climate impact due to a version upgrade is small.

[Figure]

The paper is well written, the figures are nicely prepared and the abstract provides a concise and complete summary of the paper. The structure of the paper is clear. The different intercomparison steps are well documented and sufficiently explained. In summary, the paper is a solid piece of work, which provides very valuable information for the CMIP6 community and is therefore very well suited for GMD. I recommend the paper for publications after a few minor revisions.

General comments:

There is some general issue throughout the text incl. the abstract with respect to the name of the volcanic forcing data set. To my understanding GLOSSAC and the CMIP6 Stratospheric Aerosol data set are not the same. The stratospheric aerosol data set for CMIP6 is build on the Global Satellite-based Stratospheric Aerosol Climatology (GloSSAC, Thomason et al., 2018) for the satellite area (from 1980) onwards. The version 3 (Luo, 2017) is based on GloSSAC v1.0 (Thomason et al., 2018), while the revised version v4 for Jan 1991-Dec 1994 (Luo, 2018) is based on the new data set GloSSAC v1.1 (Thomason 2018). So there exist no GLOSSAC version 4. Please check and revise the text carefully with respect to the name convention.

In the release note to version 4 of the stratospheric aerosol data set for CMIP6 (Luo, 2018), some first comparison between Stratospheric Aerosol Optical Depth and extinction of version 3 and 4 were already made with similar results as listed in section 2. This should be mentioned.

Specific comments:

Title: The title is a little bit misleading and need to be changed as the authors consider only the Post Pinatubo episode (1990 -1996) and not the full CMIP6 historical period.

Page 1, line 20 (also page 12, line 7), "can be as large as 3 C" . Maybe the authors could be more specific here and can give the exact duration and the altitude of this local maximum. If I look at figure 5, I can hardly see a temperature anomaly of 3 C. A

supplementary lon/lat figure might be helpful here to better illustrate this point.

Page 2, line 2, "an estimated 10 Tg of sulfur into the stratosphere." The S emission of Mt Pinatubo is uncertain current estimates range between 5 to 10 Tg S, see for example Timmreck et al. (2018) , p 2583.

Page 2, line 16-18, Please reformulate this sentence as it is a bit misleading. Solomon et al (2011) and Fyfe et al (2913) used an updated version of the Sato et al. (1993) data set which includes the more recent eruption.

Page 3, line 7, " an error was found " You can be more specific here and mention that it was a CLAES cloud clearing problem which affected the Pinatubo period mostly in the first months after the eruption , see "Release Notes Stratospheric Aerosol Radiative Forcing and SAD version v4.0.0 1850 - 2016 (Luo, 2018).

Page 5, line 8, Some information about the vertical resolution in the stratosphere and in the tropical tropopause region in the CanESM5 would be nice.

Page 5, line 17, Same for the EAMv1.

Page 5, line 26, One has to be careful to compare here not apples and oranges. All the cited papers (Minnis et al., 1993; Stenchikov et al., 1998; Ramachandran et al., 2000) show a decrease in net shortwave flux radiation but mention an increase in reflected shortwave radiation.

Page 8, line 8, "three realizations were performed using the EAMv1 model" As the EAMv1 model produces the QBO, I wonder about the QBO in the model. Were the QBO in different phases in the model run and how does they differ from the actual observed phase?

Page 12, line 10-11, I wonder if you had a look on possible changes in sea ice in the CanESM5?

Figure 3, The authors might think about to present the flux anomalies in the more common way with negative net short wave flux anomalies and net positive LW anomalies.

References:

Fyfe, J., Von Salzen, K., Cole, J., Gillett, N., and Vernier, J.-P.: Surface response to stratospheric aerosol changes in a coupled atmosphere–ocean model, Geophysical Research Letters, 40, 584–588, 2013.

Minnis, P., Harrison, E. F., Stowe, L. L., Gibson, G., Denn, F. M., Doelling, D., and Smith, W.: Radiative climate forcing by the Mount Pinatubo eruption, Science, 259, 1411–1415, 1993.

Luo, B., CMIP6 Stratospheric Aerosol Data Updates of version 3.0 (made available on 15 September 2017) ftp://iacftp.ethz.ch/pub_read/luo/CMIP6/ StratAerosols_CMIP6_Updates_v3.0.pdf

Luo, B.: Release Notes Stratospheric Aerosol Radiative Forcing and SAD version v4.0.0 1850 - 2016, ftp://iacftp.ethz.ch/pub_read/luo/CMIP6_SAD_radForcing_v4.0. /Release_note_v4.0.0.pdf, 2018.

Ramachandran, S., Ramaswamy, V., Stenchikov, G. L., and Robock, A.: Radiative impact of the Mount Pinatubo volcanic eruption: Lower stratospheric response, Journal of Geophysical Research: Atmospheres, 105, 24 409–24 429, 2000.

Sato, M., Hansen, J. E., McCormick, M. P., and Pollack, J. B.: Stratospheric aerosol optical depths, 1850–1990, Journal of Geophysical Research: Atmospheres, 98, 22 987–22 994, 1993.

Solomon, S., Daniel, J. S., Neely, R. R., Vernier, J.-P., Dutton, E. G., and Thomason, L. W.: The persistently variable "background" stratospheric aerosol layer and global climate change, Science, 333, 866–870, 2011

Stenchikov, G. L., Kirchner, I., Robock, A., Graf, H.-F., Antuna, J. C., Grainger, R., Lambert, A., and Thomason, L.: Radiative forcing from the 1991 Mount Pinatubo volcanic

eruption, Journal of Geophysical Research: Atmospheres, 103, 13 837–13 857, 1998.

Thomason, L. (2018). Global Space-based Stratospheric Aerosol Climatology Version 1.1 [Data set]. NASA Langley Atmospheric Science Data Center DAAC. https://doi.org/10.5067/GLOSSAC-L3-V1.1

Thomason, L. W., Ernest, N., Millán, L., Rieger, L., Bourassa, A., Vernier, J.-P., Manney, G., Luo, B., Arfeuille, F., and Peter, T.: A global space-based stratospheric aerosol climatology: 1979–2016, Earth Syst. Sci. Data, 10, 469–492, https://doi.org/10.5194/essd-10-469-2018, 2018.

Timmreck, C., Mann, G. W., Aquila, V., Hommel, R., Lee, L. A., Schmidt, A., Brühl, C., Carn, S., Chin, M., Dhomse, S. S., Diehl, T., English, J. M., Mills, M. J., Neely, R., Sheng, J., Toohey, M., and Weisenstein, D.: The Interactive Stratospheric Aerosol Model Intercomparison Project (ISA-MIP): motivation and experimental design, Geosci. Model Dev., 11, 2581–2608, https://doi.org/10.5194/gmd-11-2581-2018, 2018.
* * *

---

## Referee Comment (RC2) · Thomas Aubry (Referee) · 8 Mar 2020

Radiative forcing from stratospheric volcanic sulfate aerosol is a major cause of climate variability and a key forcing in CMIP6 historical experiment. For the satellite era, the GloSSAC dataset version 1.0 was used by all CMIP6 models to prescribe stratospheric aerosol data. However, new versions of this dataset have recently been released. The aim of the manuscript is to test whether using version 1.1 of the GloSSAC dataset instead of version 1.0 as in CMIP6 significantly affects the radiative forcing and climate response to the Mount Pinatubo 1991 eruption. Using two different Earth System Models, the authors show that differences in radiative forcing, tropospheric temperature, precipitation, ocean heat content and ENSO responses caused by the GloSSAC update are generally negligible compared to natural variability, although there are some important differences in heating rate and stratospheric temperature responses.

The manuscript is generally solid, clear, rigorous and very pleasant to read. It adresses an important question for the CMIP6 communauty and will be a very valuable contribution. I thus recomment the manuscript for publication after moderate or minor revisions. My main comment, further detailed below, is that the paper never mention the version 2.0 of the GloSSAC dataset, which I believe has stronger differences with version 1.0 than the version 1.1 tested by the authors. I think the authors should at least discuss differences between version 1.1 and 2.0, and ideally complement their results with some experiments using version 2.0.

Major comment:

The GloSSAC dataset version 2.0 was advertised at the 2019 American Geophysical Union Fall Meeting (Thomason et al. 2019, see reference list at the end) which I guess some of the authors are aware about, but this newer version is never mentioned in the manuscript. Given the main objective of the manuscript, I think the fact that a newer version exists should at the very least be discussed? I attach a plot briefly comparing global mean SAOD in v1.0, v1.1 and v2.0, which is a modified version of Figure S1 in Aubry et al. (2020) (see reference list at the end). For the Pinatubo period, v2.0 has larger SAOD than v1.0, whereas v1.1 has smaller SAOD than v1.0. Differences between v2.0 and v1.0 also tend to be more important than those between v1.1 and v1.0. Using v2.0 instead of v1.1 would thus likely change many of the results presented in this study, even though I would expect the conclusions that using any of these GloSSAC versions cause changes in the Pinatubo climate response that are small compared to the natural variability.

I recommend that the authors at least make the reader aware that a version 2.0 of the GloSSAC dataset exists and discuss differences between version 1.1 and 2.0. Adding

a SI figure similar to the nice figure 1 but showing GloSSAC v1.0, v1.1 and v2.0 would be useful. In support of this discussion, I think that the authors should reproduce figure 3 and 4 using GloSSAC v2.0. I believe this should have a relatively low computational cost given that these are 5-year AMIP simulation(s)? It would provide a first test of the differences caused by using the newest GloSSAC version in terms of radiative forcing. I believe it would additionally be a very useful contribution to further quantifying how uncertainties in stratospheric aerosol datasets - which are very challenging to build - translate in terms of radiative forcing uncertainty. Repeating the other simulations (e.g. the fully coupled CanESM simulations) with GloSSAC 2.0 would be fantastic if computational cost allows it.

I understand that the time of creation of the GloSSAC v2.0 version likely was very close to the time at which simulations for this study were conducted, and I am also not entirely sure whether the v2.0 version has been officially released (the dataset webpage still seems to mention v1.1: https://eosweb.larc.nasa.gov/project/glossac/glossac). However, given that this newer update has been advertised to the scientific community and is available (at least upon request), I believe that the authors should at the very least make the reader aware of v2.0 and discuss the differences with v1.1.

Other comments:

1) The authors mention that there is no apparent difference in El Nino Southern Oscillation (ENSO) states in the 2 years following the Pinatubo eruption, but I don't think the North Atlantic Oscillation (NAO) response is mentioned anywhere? Given that changes in stratospheric temperature response are significant and much larger in the tropics, I think it would be very valuable to show/mention whether using the new aerosol dataset affects the response of : i) the meridional temperature gradient in the stratosphere; ii) the polar vortex strength (during winter) and iii) the winter NAO phase.

2) I find the manuscript very clear, concise and pleasant to read except for the presentation of the experimental design (and to a lesser extent, for the models). The reader

discovers along the way which simulation set-up was used for which parts, with often a lack of details. For example, section 3.1 suggests that CanESM5 was used in fully-coupled mode, but then in section 4.1 it is used in AMIP mode and it is not clear how many simulations were conducted. In section 4.2, the coupled version is used and the number of ensemble member is specified, but it is not clear how initial conditions were sampled. Overall, I would prefer to see all details of experimental design in a section 3.3 clearly presenting the model setup used (for both CanESM and EAMv1) for different diagnostic, the number of ensemble members, and how initial conditions were sampled. (a table could be useful here). It would also be nice to harmonize a bit the model description, e.g. describe the model resolution in similar units (degree vs km) so that the reader can easily compare them, and give information about ability to simulate QBO for both model in their respective sections.

Specific comments:

Page 1, line 14 and 18: I believe the abstract would read a bit better if you directly mentioned that you used two different models, and then the main results from the two models.

Page 1, line 22: replace "leading to a cooling effect" by "leading to a surface cooling effect"

Page 2, line 1: I find the formulation "multiplying the impact on climate" a bit confusing; maybe replace by something like "which in turn strenghtens this surface cooling"

Page 2, line 2: as you give a range of radiative forcing you could give a range on the injected sulfur mass, which is still a major source of uncertainty.

Page 2, line 5: I would avoid expressions like "equally impressive"; maybe replace by "There was also a significant impact on oceans, "?

Page 2, line 8: More recent references you may consider to add are Stocker et al. (2019) (they use GloSSAC) and Schmidt et al. (2018)

Page 2, section 2 title: "The Stratospheric Aerosol Dataset" reads a bit funny; maybe say "The CMIP6 Stratospheric Aerosol Dataset" ?

Page 3, line 7: I think it would be neat to briefly describe what kind of error it was, in one or two sentences?

Page 3, Figure 1: this relates to my main comment, but I think having a similar figure for GloSSAC v2.0 would be nice, and I really think you have to mention and discuss this newer update.

Page 4, line 4: do you mean optical thickness? If so please clarify

Page 5, sections 3.1 and 3.2: could you give the rough vertical resolution at the altitude of the Pinatubo plume for both models?

Page 5, lines 8 and 17: to ease the model comparison, could you give the horizontal resolution either in degree or km or both?

Page 5, line 20: could you clarify whether the version you use includes this modified parameterization?

Page 5, section 3.1: could you provide in this section some information on the model capability to simulate the QBO, like you do for EAMv1 in section 3.2? You could then remove it from Page 9 line 15.

Section 4.1: it is very hard to understand which model(s) was used to conduct simulation to diagnose radiative forcing (I understand it's CanESM5 from the caption of Figure 3?). Please clarify. I think having a section 3.3 with summary of experimental design would greatly help as highlighted in one of my main comments.

Page 6, line 1: Section 3.1 gives the impression CanESM5 is used in fully-coupled mode; I'd prefer if you clarified earlier that you use it in AMIP mode to quantify changes in radiative forcing. Similarly, you say "simulations": how many? How were initial conditions sampled?

Page 6, line 5: Would shorwave/longwave be a more standard terminology than solar/thermal?

Page 6, Figure 3: even though the point of the paper is not to compare the model with observations, I think it would be neat to show some on this figure (e.g. from ERBE)

Page 7, line 3-5: just a personal preference but I think this should be in the figure caption only, not in the main text.

Page 7, line 5-9: given the reduction in heating rate is mostly in the tropic, an immediate question coming to mind whether it affects the meridional temperature gradient, winter polar vortex strength, and winter NAO response?

Page 7, Figure 4: clarify in the legend that these results are from CanESM?

Page 8, line 1-9: I wish it was clear before that coupled simulation were used to diagnose climate response and AMIP simulations for radiative forcing. Here you specify ensemble size, and you are clear about which model you use (in contrast with section 4.1), but I think you should briefly mentioned how initial conditions were sampled. (and again instead of mentioning it here I would rather have a section 3.3 devoted to the experimental design)

Page 9, line 1-2: It's nice that you show comparison with observations here.

Page 9, lines 11-13: Although the changes highlighted are small, they slightly improve consistency with observations? I think it's worth highlighting explicitely? That being said I would expect a stronger temperature response if you used the version 2.0 of GloSSAC... I really think this should be discussed.

Page 10, line 5-6: You may consider including a more recent citation for ENSO response to volcanism such as Khodri et al. (2017)

Page 10, line 6: changes in volcanic forcing before the eruption are only from January 1991 onwards, and are of magnitude smaller than 0.01 in terms of SAOD, is that correct? I find this clear shift to a La-Nina like state quite impressive given the really small changes applied for just 5 months.

Page 11, line 1-4: It's nice to comment on ENSO but given the changes in tropical stratospheric temperature you find, I am really curious to know if the winter NAO response is affected, or at least the winter polar vortex.

Page 12: Conclusions are clear and concise.

References

Thomason et al. (2019), The Global Space-based Stratospheric Aerosol Climatology: Features of v2.0, https://agu.confex.com/agu/fm19/meetingapp.cgi/Paper/502030

Aubry, T. J., Toohey, M., Marshall, L., Schmidt, A., & Jellinek, A. M. (2020). A new volcanic stratospheric sulfate aerosol forcing emulator (EVA_H): Comparison with interactive stratospheric aerosol models. Journal of Geophysical Research: Atmospheres, 125, e2019JD031303. https://doi.org/10.1029/2019JD031303

Stocker, M., Ladstädter, F., Wilhelmsen, H., & Steiner, A. K. ( 2019). Quantifying stratospheric temperature signals and climate imprints from post‐2000 volcanic eruptions. Geophysical Research Letters, 46, 12486– 12494. https://doi.org/10.1029/2019GL084396

Schmidt, A., Mills, M. J., Ghan, S., Gregory, J. M., Allan, R. P., Andrews, T., et al. ( 2018). Volcanic radiative forcing from 1979 to 2015. Journal of Geophysical Research: Atmospheres, 123, 12,491– 12,508. https://doi.org/10.1029/2018JD028776

Khodri, M., Izumo, T., Vialard, J. et al. Tropical explosive volcanic eruptions can trigger El Niño by cooling tropical Africa. Nat Commun 8, 778 (2017). https://doi.org/10.1038/s41467-017-00755-6

[Figure]

**Fig. 1.** Post-Pinatubo SAOD in GloSSAC 1.0, 1.1 and 2.0

---

## Referee Comment (RC3) · Anonymous Referee #3 · 18 Mar 2020

This manuscript presents the changes in simulated climate response (where climate is intended as temperature and precipitation) occurring when an error in the CMIP6 stratospheric aerosol forcing database in the post-Pinatubo period is corrected. The authors conclude that the correction does not significantly impact temperatures and precipitations (although there are changes in tropical stratospheric temperatures).

This manuscripts presents the results in a straightforward manner. The scientific significance is fair, in the sense that the scope of the manuscript is pretty limited, but it presents one of those results that should be documented in peer-review journals in view of the importance of the CMIP6 simulations.

[Figure]

I do not have any major comment, except for the description of the models and simulations. The descriptions of the models report very few characteristics, but there is no remarks on why these two models were chosen. It is not clear if they were the models available, or if they were chosen because their characteristics complement each other. There should be some concluding remark in the section about model description that contrast the two models against each other and make clear in which respect the results are expected or could differ, given the different characteristics. A table could also be useful, where columns report items such as "interactive SSTs" or "stratospheric chemistry".

Additionally, there is not initial description of the simulations. The simulations are introduced where they are analyzed, but it would be useful to have right after the model description a section where all simulations are presented.

---

## Author Comment (AC1) · 31 Jul 2020

The authors would like to thank reviewer #1 for the thoughtful comments. Their suggestions have helped to clarify important model parameters as well as experimental results.

**1** General Comments**

There is some general issue throughout the text incl. the abstract with respect to the name of the volcanic forcing data set. To my understanding GLOSSAC and the CMIP6 Stratospheric Aerosol data set are not the same. The stratospheric aerosol data set for CMIP6 is build on the Global Satellite-based Stratospheric Aerosol Climatology (GloSSAC, Thomason et al., 2018) for the satellite area (from 1980) onwards. The version 3 (Luo, 2017) is based on GloSSAC v1.0 (Thomason et al., 2018), while the revised version v4 for Jan 1991-Dec 1994 (Luo, 2018) is based on the new data set GloSSAC v1.1 (Thomason 2018). So there exist no GLOSSAC version 4. Please check and revise the text carefully with respect to the name convention.

Lines 6-10 of the abstract have been updated to:

"To improve this situation for CMIP6 a two step process was undertaken. First, a combined stratospheric aerosol dataset, the Global Space-based Stratospheric Aerosol Climatology, GloSSAC, was constructed. Next, GloSSAC, along with information from ice-cores and sun photometers, was used to generate aerosol distributions, characteristics and optical properties to construct a consistent stratospheric aerosol forcing dataset for models participating in CMIP6."

In the release note to version 4 of the stratospheric aerosol data set for CMIP6 (Luo, 2018), some first comparison between Stratospheric Aerosol Optical Depth and extinction of version 3 and 4 were already made with similar results as listed in section 2. This should be mentioned.

Thank you, that should be included and this is now discussed on Page 3 Lines 16-17. *"Luo et al., (2018) show the magnitude of these changes at several latitude bins and times for 1020 nm, and the following analysis expands on this at 550 nm in the context of this paper."*
**2 Specific Comments**

Title: The title is a little bit misleading and need to be changed as the authors consider only the Post Pinatubo episode (1990 -1996) and not the full CMIP6 historical period.

Changed to "Quantifying CanESM5 and EAMv1 sensitivities to Mt. Pinatubo volcanic forcing for the CMIP6 historical experiment"

Page 1, line 20 (also page 12, line 7), "can be as large as 3 C". Maybe the authors could be more specific here and can give the exact duration and the altitude of this local maximum. If I look at figure 5, I can hardly see a temperature anomaly of 3C. A supplementary lon/lat figure might be helpful here to better illustrate this point.

Thank you, an additional latitude-time figure (Figure 6 in the paper also attached here) has been added to clarify the temperature anomalies and discussed on page 10 Lines 10-13. Hopefully this new figure, along with Figure 5, help clarify the spatial extent and magnitude of the anomalies.

Page 2, line 2, "an estimated 10 Tg of sulfur into the stratosphere." The S emission of Mt Pinatubo is uncertain current estimates range between 5 to 10 Tg S, see for example Timmreck et al. (2018), p 2583.

Revised to "the 1991 eruption of Mt. Pinatubo injected an estimated 5-10 Tg of sulfur into the stratosphere (Guo et al., 2014, English et al., 2013, Dhomse et al., 2014, Timmreck et al., 2018)" on Page 2 Line 2.

Page 2, line 16-18, Please reformulate this sentence as it is a bit misleading. Solomon et al (2011) and Fyfe et al (2013) used an updated version of the Sato et al. (1993) data set which includes the more recent eruption.

We don't think Solomon et al., (2011) used the most recent version of Sato et al., but
instead used the *Vernier et al.*, (2011) climatology derived from CALIPSO measurements to extend measurements post-2000. The updated Sato dataset was published in December 2012, and used OSIRIS data as opposed to CALIPSO, so these datasets will differ somewhat. While *Fyfe et al.*, (2013) used the updated Sato climatology, they did so only until 1993 (1998 was also tested), at which point they transitioned to the Vernier dataset.

Page 3, line 7, " an error was found " You can be more specific here and mention that it was a CLAES cloud clearing problem which affected the Pinatubo period mostly in the first months after the eruption , see "Release Notes Stratospheric Aerosol Radiative Forcing and SAD version v4.0.0 1850 - 2016 (Luo, 2018).

Thank you, updated with suggested explanation on Page 3 Lines 7-8.

**Page 5, line 8, Some information about the vertical resolution in the stratosphere and in the tropical tropopause region in the CanESM5 would be nice**

The vertical resolution is approximately 1-2km in the lowermost stratosphere. This has been added to the CanESM5 and EAMv1 model description on Page 7 Line 6.

**Page 5, line 17, Same for the EAMv1.**

The vertical resolution is approximately 1-2km in the lowermost stratosphere. This has been added to the EAMv1 model description on Page 7 Line 16.

Page 5, line 26, One has to be careful to compare here not apples and oranges. All the cited papers (Minnis et al., 1993; Stenchikov et al., 1998; Ramachandran et al., 2000) show a decrease in net shortwave flux radiation but mention an increase in reflected shortwave radiation.

Thank you, clarified to "and increases in reflected radiation at the top of the atmosphere" on Page 7 Line 25.

Page 8, line 8, "three realizations were performed using the EAMv1 model" As
the EAMv1 model produces the QBO, I wonder about the QBO in the model. Were the QBO in different phases in the model run and how does they differ from the actual observed phase?

The QBOs were in different phases during the eruption, although none matched the observed phasing precisely. The attached figure shows the QBO Index (as calculated from Christy and Drouilhet (1994) for the three EAM simulations and RSS observations. This is now briefly discussed on Page 7 Lines 17-18.

Christy, J. R., & Drouilhet Jr, S. J. (1994). Variability in daily, zonal mean lower-stratospheric temperatures. Journal of climate, 7(1), 106-120.

**Page 12, line 10-11, I wonder if you had a look on possible changes in sea ice in the CanESM5?**

We did not look at sea ice specifically, but both the ocean heat content, and temperature outside of the tropics remain unchanged between version. The now included Figure 6 on Page 11 shows this more clearly and discussed on Page 10 Lines 13-15.

Figure 3, The authors might think about to present the flux anomalies in the more common way with negative net short wave flux anomalies and net positive LW anomalies.

Switched throughout to the more common convention.

---

## Author Comment (AC2) · 31 Jul 2020

The authors would like to thank reviewer #2 for the thorough review. In particular, mention of the upcoming of GloSSAC v2 is an important point, and the notes on model and experimental description have hopefully helped to clarify results.

**1 Major Comments**

The GloSSAC dataset version 2.0 was advertised at the 2019 American Geophysical Union Fall Meeting (Thomason et al. 2019, see reference list at the end) which I guess some of the authors are aware about, but this newer version is never mentioned in the manuscript. Given the main objective of the manuscript, I think the fact that a newer version exists should at the very least be discussed? I attach a plot briefly comparing global mean SAOD in v1.0, v1.1 and v2.0, which is a modified version of Figure S1 in Aubry et al. (2020) (see reference list at the end). For the Pinatubo period, v2.0 has larger SAOD than v1.0, whereas v1.1 has smaller SAOD than v1.0. Differences between v2.0 and v1.0 also tend to be more important than those between v1.1 and v1.0. Using v2.0 instead of v1.1 would thus likely change many of the results presented in this study, even though I would expect the conclusions that using any of these GloSSAC versions cause changes in the Pinatubo climate response that are small compared to the natural variability.

I recommend that the authors at least make the reader aware that a version 2.0 of the GloSSAC dataset exists and discuss differences between version 1.1 and 2.0. Adding a SI figure similar to the nice figure 1 but showing GloSSAC v1.0, v1.1 and v2.0 would be useful. In support of this discussion, I think that the authors should reproduce figure 3 and 4 using GloSSAC v2.0. I believe this should have a relatively low computational cost given that these are 5-year AMIP simulation(s)? It would provide a first test of the differences caused by using the newest GloSSAC version in terms of radiative forcing. I believe it would additionally be a very useful contribution to further quantifying how uncertainties in stratospheric aerosol datasets - which are very challenging to build - translate in terms of radiative forcing uncertainty. Repeating the other simulations (e.g. the fully coupled CanESM simulations) with GloSSAC 2.0
would be fantastic if computational cost allows it.

I understand that the time of creation of the GloSSAC v2.0 version likely was very close to the time at which simulations for this study were conducted, and I am also not entirely sure whether the v2.0 version has been officially released (the dataset webpage still seems to mention v1.1: https://eosweb.larc.nasa.gov/project/glossac/glossac). However, given that this newer update has been advertised to the scientific community and is available (at least upon request), I believe that the authors should at the very least make the reader aware of v2.0 and discuss the differences with v1.1.

Thank you, and we agree that the new GloSSAC v2 will be an important addition to the climate record and should be noted. We now discuss version 2 with updated references and include a supplemental figure comparing the glossac versions in the same manor as the forcing datasets (also attached here). Please see Page 5 lines 1-7 and Supplemental Figure S1 of the revised manuscript.

A more complete analysis including radiative forcing differences would indeed be very interesting; however, at the time of writing the optical properties and climatologies that are derived from GloSSAC v2 (and used as the inputs to the climate models) have not yet been produced. Conversion from GloSSAC to climate forcing files is a detailed process with many variables, making any use of v2 in climate studies a comparison of both the GloSSAC version and conversion steps needed to translate the GloSSAC data into climate model input. Additionally, we have updated the manuscript to hopefully clarify the difference between the underlying GloSSAC dataset and the derived CMIP climatology.
**2 Other Comments**

The authors mention that there is no apparent difference in El Nino Southern Oscillation (ENSO) states in the 2 years following the Pinatubo eruption, but I don't think the North Atlantic Oscillation (NAO) response is mentioned anywhere? Given that changes in stratospheric temperature response are significant and much larger in the tropics, I think it would be very valuable to show/mention whether using the new aerosol dataset affects the response of : i) the meridional temperature gradient in the stratosphere; ii) the polar vortex strength (during winter) and iii) the winter NAO phase.

i) A figure showing the meridional temperature gradient at 4 pressure levels is attached. Dotted regions indicate where no statistically significant change has occurred. Significant differences in temperature gradients are not present outside of the tropics or at levels other than 20, 30, and 50 hPa.

ii)Although the polar vortex strength was not looked at specifically, the temperature gradient outside of 30°S-30°N is unaffected by the changes in aerosol, so no changes to the polar vortex are expected. This is now noted on Page 10 Line 13. Further investigation into the effects of additional parameters would certainly be interesting, but likely require larger ensembles as no significant effects are noticeable in either temperature outside of the tropics, ocean heat content, ENSO or NAO (see next point) in the current ensembles.

iii)NAO was investigated and no significant change was found between version 3 and version 4 of the forcing datasets. The attached figure shows the NAO index between 1990 and 1996. Lines indicate ensemble means and shaded areas indicate 10 and 90th percentiles. That no change is seen in the NAO index is now also mention on Page 13 Lines 1-2.
I find the manuscript very clear, concise and pleasant to read except for the presentation of the experimental design (and to a lesser extent, for the models). The reader discovers along the way which simulation set-up was used for which parts, with often a lack of details. For example, section 3.1 suggests that CanESM5 was used in fully coupled mode, but then in section 4.1 it is used in AMIP mode and it is not clear how many simulations were conducted. In section 4.2, the coupled version is used and the number of ensemble member is specified, but it is not clear how initial conditions were sampled. Overall, I would prefer to see all details of experimental design in a section 3.3 clearly presenting the model setup used (for both CanESM and EAMv1) for different diagnostic, the number of ensemble members, and how initial conditions were sampled. (a table could be useful here). It would also be nice to harmonize a bit the model description, e.g. describe the model resolution in similar units (degree vs km) so that the reader can easily compare them, and give information about ability to simulate QBO for both model in their respective sections.

Thank you, section 3 has been revised to include more detail on experiment setup as well as provide a more complete overview of the models. See Page 6, lines 1-23 and Table 1.

**3** Specific Comments**

Page 1, line 14 and 18: I believe the abstract would read a bit better if you directly mentioned that you used two different models, and then the main results from the two models.

Page 1 Lines 14-20 have been updated to "This study uses two models, the Canadian Earth System Model version 5 (CanESM5), and Energy Exascale Earth System Model (E3SM) Atmosphere Model version 1
(EAMv1) to estimate the difference in instantaneous radiative forcing in simulated post-Pinatubo climate response when using version 4 instead of version 3. Differences in temperature, precipitation, and radiative forcings are generally found to be small compared to internal variability. An exception to this is differences in monthly temperature anomalies near 24 km altitude in the tropics, which can be as large as  $3^{\circ}C$  following the eruption of Mt. Pinatubo."

Page 1, line 22: replace "leading to a cooling effect" by "leading to a surface cooling effect"

Thank you, updated on Page 1 Line 22.

Page 2, line 1: I find the formulation "multiplying the impact on climate" a bit confusing; maybe replace by something like "which in turn strengthens this surface cooling"

Changed as suggested on Page 1 Line 23

Page 2, line 2: as you give a range of radiative forcing you could give a range on the injected sulfur mass, which is still a major source of uncertainty.

Updated on Page 2 line to

"For example, the 1991 eruption of Mt. Pinatubo injected an estimated 5-10 Tg of sulfur into the stratosphere (Guo et al., 2004, English et al., 2013, Dhomse et al., 2014)"

Page 2, line 5: I would avoid expressions like "equally impressive"; maybe replace by "There was also a significant impact on oceans, "?

Updated Page 2 line 4 as suggested.

Page 2, line 8: More recent references you may consider to add are Stocker et al. (2019) (they use GloSSAC) and Schmidt et al. (2018)

Thank you, added on Page 2 Lines 8-9.

GMDD
Page 2, section 2 title: "The Stratospheric Aerosol Dataset" reads a bit funny; maybe say "The CMIP6 Stratospheric Aerosol Dataset" ?

Updated on Page 2 Line 26.

Page 3, line 7: I think it would be neat to briefly describe what kind of error it was, in one or two sentences?

Agreed, this has been added on Page 3 Lines 7-8.

Page 3, Figure 1: this relates to my main comment, but I think having a similar figure for GloSSAC v2.0 would be nice, and I really think you have to mention and discuss this newer update.

Thank you, and we agree that GloSSAC v2 should be mentioned. Supplemental Figure S1 is now included (and attached below) and v2 is discussed on Page 5 Lines 1-7.

**Page 4, line 4: do you mean optical thickness? If so please clarify**

Thank you, corrected on Page 4 Line 4/5.

**Page 5, sections 3.1 and 3.2: could you give the rough vertical resolution at the altitude of the Pinatubo plume for both models?**

Vertical resolutions of CanESM5 (1.5km) and EAMv1 (1-2km) in the lowermost stratosphere have been added to the model description sections. See Page 7 Lines 6 and 16 respectively.

**Page 5, lines 8 and 17: to ease the model comparison, could you give the horizontal resolution either in degree or km or both?**

Horizontal model resolution now specified in both km and degrees on Page 7 lines 5 and 15.

Page 5, line 20: could you clarify whether the version you use includes this modified parameterization?
The parameterization was not included in this version. This has been clarified in the manuscript as: "While the QBO can be greatly improved by modifying parameterized convectively generated gravity waves (Richter et al., 2019) this is not included in the current simulations.", on Page 7 lines 18-19.

Page 5, section 3.1: could you provide in this section some information on the model capability to simulate the QBO, like you do for EAMv1 in section 3.2? You could then remove it from Page 9 line 15.

Added to CanESM5 description on Page 7 line 8 and removed from Page 9.

Section 4.1: it is very hard to understand which model(s) was used to conduct simulation to diagnose radiative forcing (I understand it's CanESM5 from the caption of Figure 3?). Please clarify. I think having a section 3.3 with summary of experimental design would greatly help as highlighted in one of my main comments.

Thank you, hopefully this is addressed in the updated section 3 (See Page 6, lines 1-23) and Table 1.

Page 6, line 1: Section 3.1 gives the impression CanESM5 is used in fullycoupled mode; I'd prefer if you clarified earlier that you use it in AMIP mode to quantify changes in radiative forcing. Similarly, you say "simulations": how many? How were initial conditions sampled?

See previous note.

Page 6, line 5: Would shortwave/longwave be a more standard terminology than solar/thermal?

Perhaps, the cited literature mentions both shortwave/infrared/longwave and solar/thermal radiation, so we have chosen to go with the latter convention.

Page 6, Figure 3: even though the point of the paper is not to compare the model

GMDD
**with observations, I think it would be neat to show some on this figure (e.g. from ERBE)**

We agree this would be an interesting comparison. However, the volcanic signal is clear in this analysis only because the radiation calculation is ran with and without stratospheric aerosols, so is not possible with observational datasets.

**Page 7, line 3-5: just a personal preference but I think this should be in the figure caption only, not in the main text.**

Agreed, removed from main text.

**Page 7, line 5-9: given the reduction in heating rate is mostly in the tropic, an immediate question coming to mind whether it affects the meridional temperature gradient, winter polar vortex strength, and winter NAO response?**

See note above for more details, but the meridional temperature gradient is also unaffected outside of the tropics. This is now noted on Page 10 Line 13.

**Page 7, Figure 4: clarify in the legend that these results are from CanESM?**

Thank you, now clarified in the caption as *"instantaneous heating rates from CanESM5 model runs"*.

Page 8, line 1-9: I wish it was clear before that coupled simulation were used to diagnose climate response and AMIP simulations for radiative forcing. Here you specify ensemble size, and you are clear about which model you use (in contrast with section 4.1), but I think you should briefly mentioned how initial conditions were sampled. (and again instead of mentioning it here I would rather have a section 3.3 devoted to the experimental design)

Agreed, Hopefully Section 3 now more clearly explains experimental setup and what simulations were used for which analysis.

Page 9, lines 11-13: Although the changes highlighted are small, they slightly

GMDD
improve consistency with observations? I think it's worth highlighting explicitly? That being said I would expect a stronger temperature response if you used the version 2.0 of GloSSAC... I really think this should be discussed.

We certainly agree it's an interesting question, and now discuss the changes to GloS-SAC v2. However, the assumptions involved in translating from GloSSAC extinction values to optical properties integrated over model bands are not trivial, and make this difficult to answer until that process is performed in a comparable way to that done for CMIP6 models. At the time of writing we are unaware of when that process will be performed, or of a paper describing the details which are required to reproduce it.

**Page 10, line 5-6: You may consider including a more recent citation for ENSO response to volcanism such as Khodri et al. (2017)**

Thank you, this reference has been added on Page 12 Line 5.

Page 10, line 6: changes in volcanic forcing before the eruption are only from January 1991 onwards, and are of magnitude smaller than 0.01 in terms of SAOD, is that correct? I find this clear shift to a La-Nina like state quite impressive given the really small changes applied for just 5 months

That is correct that the changes are much small. However, the shift to La-Nina states is also small and not clear statistically with an ensemble size of 15. The attached figure shows the ENSO state of the 15 ensemble members for June 1991. Therefore, We don't think there is any reason to suspect the slight change in ENSO phasing is anything other internal variability.

**Page 11, line 1-4: It's nice to comment on ENSO but given the changes in tropical stratospheric temperature you find, I am really curious to know if the winter NAO response is affected, or at least the winter polar vortex.**

See above plot and comment for more details, but the NAO and temperature of the higher latitudes is unaffected by the changes in aerosol.

---

## Author Comment (AC3) · 31 Jul 2020

The authors would like to thank reviewer #3 for their comments. We have updated the model and experimental descriptions as per your suggestions and think this made for a much clearer manuscript.

[Figure]

**1 General Comments**

**This manuscript presents the changes in simulated climate response (where climate is intended as temperature and precipitation) occurring when an error in the CMIP6 stratospheric aerosol forcing database in the post-Pinatubo period is corrected. The authors conclude that the correction does not significantly impact temperatures and precipitations (although there are changes in tropical stratospheric temperatures). This manuscripts presents the results in a straightforward manner. The scientific significance is fair, in the sense that the scope of the manuscript is pretty limited, but it presents one of those results that should be documented in peer-review journals in view of the importance of the CMIP6 simulations.**

**I do not have any major comment, except for the description of the models and simulations. The descriptions of the models report very few characteristics, but there is no remarks on why these two models were chosen. It is not clear if they were the models available, or if they were chosen because their characteristics complement each other. There should be some concluding remark in the section about model description that contrast the two models against each other and make clear in which respect the results are expected or could differ, given the different characteristics. A table could also be useful, where columns report items such as "interactive SSTs" or "stratospheric chemistry".**

**Additionally, there is not initial description of the simulations. The simulations are introduced where they are analyzed, but it would be useful to have right after the model description a section where all simulations are presented.**

Thank you, Section 3 has been updated with a description of the simulations and why the particular models were chosen has been added. Please see Page 6 lines 1-21,

and table 1 in the revised manuscript.